# Molecular and Pathogenic Characterization of *Fusarium* Species Associated with Corm Rot Disease in Saffron from China

**DOI:** 10.3390/jof8050515

**Published:** 2022-05-17

**Authors:** Seyed Ali Mirghasempour, David J. Studholme, Weiliang Chen, Weidong Zhu, Bizeng Mao

**Affiliations:** 1Institute of Biotechnology, Zhejiang University, Hangzhou 310058, China; ali_mirghasempor@yahoo.com (S.A.M.); chenwl@zju.edu.cn (W.C.); 2Biosciences, College of Life and Environmental Sciences, University of Exeter, Exeter EX4 4QD, UK; d.j.studholme@exeter.ac.uk; 3Zhejiang Shouxiangu Pharmaceutical Co., Ltd., Wuyi 321200, China; zwd13516905199@163.com

**Keywords:** *Crocus sativus*, corm rot, *Fusarium annulatum*, *F. commune*, MLSA

## Abstract

Saffron (*Crocus sativus* L.) is a commercial spice crop well-known throughout the world, valued for culinary, colorant, and pharmaceutical purposes. In China, *Fusarium nirenbergiae* was detected as causative agent of saffron corm rot, the most pervasive disease for the first time in 2020. In the present study, 261 *Fusarium*-like isolates were recovered from 120 rotted corms in four saffron producing fields at Zhejiang, Shanghai, and Yunnan provinces, China, in 2021. A combination of morpho-cultural features and multilocus sequence analysis (MLSA) of the concatenated *rpb2* (DNA-directed RNA polymerase II largest subunit) and *tef1* (translation elongation factor 1-α) partial sequences showed that the isolates from saffron belong to *Fusarium nirenbergiae* as well as *F. commune*, and *F. annulatum* with isolation frequencies of 58.2%, 26.8%, and 14.9%, respectively. Notably, *F. commune* was more prevalent than *F. annulatum* in the collected samples. Pathogenicity tests confirmed that both species were pathogenic on saffron corm. This is the first report of *F. annulatum* and *F. commune* causing corm rot of saffron, globally. Outcomes of the current research demonstrate that *Fusarium* spp. associated with saffron corm rot are more diverse than previously reported. Furthermore, some plants were infected by two or more *Fusarium* species. Our findings broaden knowledge about *Fusarium* spp. that inflict corm rot and assist the development of control measures.

## 1. Introduction

*Crocus sativus* (*Iridaceae*) is a bulbous perennial herb that is cultivated in warm temperate and subtropical countries throughout the world. Its vivid crimson stigmata and styles of the flowers are used as a desirable condiment, dye, aroma, antioxidant, and in medicine [1,2]. Saffron is a triploid, male-sterile plant incapable of developing fertile seeds for reproduction; this plant can be propagated only vegetatively by its corm [3]. The major saffron varieties include Aquilla, Crème, Kashmiri, Mongra, Organic, Persian, Spanish, and Superior [3,4,5]. Nevertheless, only the “Fanhong Hua1” cultivar is widely cultivated in China. This plant is widely used as an important herbal medicine crop in China with a total cultivating area of nearly 600 hectares.

Growth of a saffron crop is hampered by biotic agents, such as insects, fungi, viruses, and bacteria. Various pathogens affect corm, including *Aspergillus niger*, *A. terreus*, *A. flavus*, *A. flavipes*, *Bacillus croci*, *Burkholderia gladioli*, *Fusarium* spp., *Macrophomina phaseolina*, *Mucor* sp., *Penicillium solitum*, *P. cyclopium*, *P. corymbiferum*, *Phoma crocophila*, *Pythium* spp., *Rhizoctonia crocorum*, *R. violacea*, *Rhizopus nigricans*, *Sclerotium rolfsii*, *Stromatinia gladioli*, and *Uromyces croci* [6,7,8,9,10]. Among them, *Fusarium* corm rot is one of the most destructive belowground fungal diseases in saffron that causes significant economic losses. The disease was first described in Japan [6], then was detected in China, India, Iran, Italy, and Spain. Previous investigators have reported that corm rot is caused by various *F. oxysporum formae speciales* such, as *gladioli*, *iridiacearum*, and *saffrani* [7,8,9]. Recently, Mirghasempour et al. [10] identified *F. nirenbergiae* as the predominant agent of corm rot in China.

*Fusarium* is a ubiquitous genus of filamentous fungi with varying morphological, physiological, and ecological features that causes economic damage on agricultural products and includes opportunistic human pathogens [11,12]. However, there are non-pathogenic strains of *Fusarium* that are used in plant protection [12]. In addition, these fungi are able to synthesize phytotoxins and mycotoxins as secondary metabolites, which can play an important role in the pathogenesis [13,14]. To date, more than 400 phylogenetically distinct species in 23 monophyletic species complexes are included in the genus *Fusarium*. Several monotypic lineages have been characterized of which members of the *F. solani* species complex (FSSC), *F. oxysporum* species complex (FOSC), *F. fujikuroi* species complex (FFSC), *F. incarnatum-equiseti* species complex (FIESC), and *F. sambucinum* species complex (FSSC) cause considerable diseases in plants [11,12,15,16,17,18,19]. Some *Fusarium* species are considered to be hemibiotrophs capable of switching to necrotrophs depending on the host and environmental conditions [14,20]. Furthermore, it has been difficult to distinguish closely related *Fusarium* species, due to inter-/intra-species overlap and inconsistency in morphological traits. By applying a polyphasic taxonomic approach that combines morphological observations with DNA fingerprinting, and multilocus phylogenetic analysis (MLSA), numerous species have been delineated within *Fusarium* spp. complexes, leading to significant improvement in the *Fusarium* classification system [11,12,13,19,21,22].

Limited information is available on the genetic diversity, phylogenetic relationships, and epidemiology of *Fusarium* species causing saffron corm rot in China and elsewhere in the world. Therefore, the aim of the current research was to identify and characterize *Fusarium* spp. associated with the disease on *Crocus sativus* through pathogenicity tests, morphological data, and molecular methods.

## 2. Materials and Methods

### 2.1. Fungal Isolation and Morphological Characterization

Saffron plants (120) with rotted corms were sampled from four saffron cultivation areas in Zhejiang (Jiande and Wuyi cities), Yunnan (Shangri-la city), and Shanghai (Chong Ming Dao Island) provinces, China (Table 1). Excised symptomatic tissues consisting of diseased and healthy parts were surface-sterilized with a 2% solution of sodium hypochlorite (0.1% active ingredient of chlorine) for 1 min and 75% ethanol for 30 s. The samples were then washed thrice with sterile distilled water, air-dried on the sterile filter papers under aseptic conditions, and finally plated onto Potato Dextrose Agar (PDA) plates, which were incubated in the dark at 25 °C. Purified isolates were obtained by hyphal tipping; then, they were sub-cultured on PDA and synthetic nutrient-poor agar (SNA) media [11,23]. Morphological characteristics of fungal colonies were meticulously examined under a Nikon Eclipse microscope (Tokyo, Japan).

### 2.2. DNA Sequencing and Molecular Phylogenetic Analysis

The mycelium of 7-day-old isolates was harvested from PDA by scraping the colony surface, freezing the mycelium in liquid nitrogen, and then grounding with a sterile mortar and pestle. The genomic DNA of thirteen representative isolates was extracted using a Plant Genomic DNA kit (Tiangen, China) according to the company’s protocols. Portions of nuclear translation elongation factor 1-alpha (*tef1*), second largest subunit of RNA polymerase II gene (*rpb2*), and internal transcribed spacer (*ITS*) genes were amplified from the thirteen representative isolates using the primers EF-1/EF-2, RPB2-5f2/RPB2-7cr, and ITS1/ITS4, respectively [10,11]. Polymerase chain reaction (PCR) was conducted with 1 μL forward primer (10 μM), 1 μL reverse primer (10 μM), 0.5 μL DNTPs (10 mM), 12.5 μL of 2× Rapid Taq Master Mix (Vazyme, Nanjing, China), 1 µL of genomic DNA, and 9.5 μL of DNase-free water. The PCR program consisted of an initial denaturation at 95 °C for 2 min, 35 cycles of denaturation at 95 °C for 30 s, annealing for 30 s at 55 °C, and an extension at 72 °C for 2 min, followed by a final extension (72 °C, 10 min). The amplified products were visualized on a 1% agarose gel and then sequenced in both directions to ensure high accuracy by Sangon Biotech Co., Ltd. (Shanghai, China). All sequences obtained in this study were deposited in GenBank and the accession numbers are included in Table 2. We performed BLASTN searches via the NCBI BLAST web portal (available online: https://blast.ncbi.nlm.nih.gov/Blast.cgi, accessed on 15 August 2021) to gather related sequences for inclusion in phylogenetic analysis. BLASTN searches were performed against the Nucleotide collection (nr/nt) and included searches were restricted to sequences from type material. After finalizing the multiple alignments by MUSCLE [24] for individual and concatenated loci (*rpb2* + *tef1*), the Kimura two-parameter model assuming a discrete gamma distribution and invariant sites (K2 + G + I) was used to estimate the best substitution models. Phylogenetic inference was obtained using maximum likelihood (ML) in MEGA X (Molecular Evolutionary Genetics Analysis version 10.2.4 [25]. Branch stability was estimated with 1000 bootstrap replicates. We included sequences from type strains of *Fusarium* species initially identified as closely related to our sequences (Table 3) by preliminary BLAST searches.

### 2.3. Pathogenicity Studies

An in vitro virulence test was conducted to evaluate the ability of thirteen representative isolates of *Fusarium* species to colonize saffron and induce rot symptoms with two methods. In the first method, inoculation of corms was done as described by Palmero et al. [7] with slight modifications. Briefly, the intact bulbs were surface-disinfested with 5% sodium hypochlorite solution (0.26% active ingredient of chlorine) for 10 min followed by 75% ethanol for 1 min and then rinsed three times in sterile water. Each isolate was cultured on PDB (potato dextrose broth) medium for 7 days at 25 °C under shaking (150 rpm) conditions. Conidial suspensions were filtered through three layers of sterilized gauze and centrifuged at 8000 rpm for 10 min. To remove the PDB, the conidial pellet was washed three times in sterile distilled water. The spores were resuspended to a final concentration of 1 × 10^6^ conidia/mL for inoculation. To completely absorb pathogens, the corms were submerged in the spore suspension and then planted in an aseptic substrate (black/white peat, perlite, vermiculite; 2:1:1) and maintained for 21 days under controlled conditions in a growth chamber with 12 h photoperiod, 23 ± 2 °C, and 70% relative humidity. The controls were inoculated with sterile water. In the second method, the mycelial plugs (5 mm diameter) were placed onto sterilized bulbs and wrapped with Parafilm, then incubated in the same condition as mentioned above for 14 days. Non-colonized PDA discs were used as negative controls. The experiments were repeated twice. The disease progression on inoculated plants was inspected daily for up to three weeks and visual observations recorded. Koch’s postulates were fulfilled by reisolating and identifying the fungal isolates from symptomatic corms.

## 3. Results

### 3.1. Field Survey, Disease Symptoms, and Pathogen Isolations

In 2021, a total of 120 diseased *Crocus sativus* plants exhibiting corm rot, leaf chlorosis, and wilted shoots were collected from four saffron-growing areas in China (Figure 1). Overall, 261 *Fusarium*-like isolates were obtained from symptomatic corms tissues (Table 1). Most isolates were obtained from individual rotted area on the corms. Based on their colony characteristics as well as molecular methods, the isolated fungi were identified as three species of *Fusarium*, namely *F. annulatum*, *F. commune*, and *F. nirenbergiae*. The isolation frequency of *F. nirenbergiae* (58.2%) was greater than *F. commune* (26.8%), and *F. annulatum* (14.9%). The ability of *F. nirenbergiae* to cause corm rot has been previously established [10], and was therefore not further examined in the current study. Here, we focused on thirteen representative isolates of *F. annulatum*, and *F. commune*, chosen based on geographical regions and identified using sequences of *tef1* and *ITS* (internal transcribed spacer). The presence of one, two, and occasionally three diverse *Fusarium* species was confirmed from some samples based on the *tef1* and *ITS* sequences.

### 3.2. Morphological Identification

Morphological features of the thirteen representative isolates recovered from symptomatic corms were consistent with the morphological descriptions of *F. annulatum* and *F. commune* [19,26]. On PDA, the abundant aerial mycelium of *F. annulatum* was pinkish white to maroon. The macroconidia were fusiform, cylindrical, narrow, straight to slightly curved with 3–6 septa. No chlamydospores were found. The microconidia were typically aseptate, club-shaped each with curved apical cell; these developed on monophialides and polyphialides (Figure 2). On PDA, *Fusarium commune* formed densely floccose to fluffy aerial mycelium with white to pale lilac colony. The macroconidia were banana-shaped, with 3–5 septa, forming sporodochia. The microconidia were cylindrical to ovate-oblong, primarily aseptate, and nesting on aerial polyphialides or, less often, on monophialides. Spherical, intercalary, or terminal chlamydospores were produced singly or in pairs (Figure 2).

### 3.3. Molecular Characterization and Phylogeny

The PCR amplification of *tef1*, *ITS*, and *rpb2* regions for isolates from saffron generated 647, 542, and 941 bp fragments, respectively. BLASTN of *ITS* sequences indicated close similarity with *Fusarium* species but provided insufficient resolution to identify the species. On the other hand, the *tef1* and *rpb2* indicated that five of the isolates were closely related to the type strain of *F. annulatum* while the other eight isolates were closely related to *F. commune*, which supported our preliminary morphological identification of these isolates. Isolates WFA10, WFA18, JFA12, JFA15, and SFA4 showed close sequence similarity with *F. annulatum* at both the *tef1* and *rpb2* loci. These isolates shared 99.05% identity with *F. annulatum* type strain CBS258.54 [19,27] at the *tef1* locus and 99.37% to this type-strain at the *rpb2* locus. Isolates YFC2, YFC5, JFC1, JFC7, SFC6, SFC20, WFC3, and WFC8 shared 97.45% identity with the type strain of *F. commune* CBS 110090 [11] at the *rpb2* locus. No *tef1* sequence is available for the *F. commune* type strain; however, the *tef1* of the four isolates shared 100% identity with *tef1* sequences of other strains designated as *F. commune* (e.g., GenBank accessions MG888467.1, MF150040.1, MW589548.1, MT313846.1). To further elucidate and illustrate the phylogenetic relations, we generated phylogenetic trees based on *tef1* and *rpb2*, including sequences of isolates recovered from saffron, *Fusarium* type strains plus non-type strains of *F. commune*. These trees further supported the identification of eight isolates as *F. commune* and five isolates as *F. annulatum* (Appendix A). The phylogenetic tree, based on the concatenation of two genes (*rpb2* + *tef1*) spanning 1476 nucleotides among 80 ingroup strains, included three main clades corresponding to FOSC, FFSC, and *F. commune*. The MLSA tree illustrated that isolates collected from saffron in the present study clustered strongly with *F. annulatum* [19] and *F. commune* type strains [26,28] with bootstrap values 95% and 100%, respectively (Figure 3). The topology of the multilocus tree was similar to the phylogenetic trees constructed from the individual genes (Appendix A). Moreover, *F. nirenbergiae* JD1, which is a major causative pathogen of saffron rot in China, also falls unambiguously within the *F. nirenbergiae* clade (88% bootstrap value), which belongs to FOSC (Figure 3).

### 3.4. Pathogenicity Assays

Conidial and mycelial inoculation with thirteen representative isolates resulted in rotting and wilting symptoms in corms three weeks after inoculation under laboratory conditions. These symptoms were similar to field observations, whereas no symptoms were observed on control plants. The visual assessments indicated that the disease development in the corms inoculated by mycelial plugs was faster than conidial inoculation. The fungal isolates were reisolated from the inoculated corms and identified using *tef1* locus to fulfill Koch’s postulates (Figure 4). As such, it was verified that *F. annulatum* and *F. commune* were capable of causing rot in corms. While the virulence of *F. commune* isolates was visually greater than *F. annulatum* isolates, the corm rot symptoms incited by each species were not distinguishable.

## 4. Discussion

Saffron corm rot is the most challenging disease of saffron with a high incidence in the Chinese provinces Shanghai, Yunnan, and Zhejiang. *Fusarium* species are among the most severe pathogens that affect a broad range of crops worldwide. We previously established that *F. nirenbergiae* is a causative agent of corm rot on saffron [10]. In the current study, we isolated and identified two further species from saffron growing regions, namely *Fusarium annulatum* and *F. commune*, based on morphological criteria and multilocus (two-gene) phylogenetic analyses. Isolates identified as these two species were clearly distinct from *F. nirenbergiae* isolates previously described [10]. The establishment of Koch’s postulates indicated that *F. annulatum* and *F. commune* isolates were pathogenic to saffron. However, they have a slight variation in virulence. The occurrence of these pathogens in different locations of China suggests that infected corms may serve as a source of inoculum for *C. sativus* infection.

*Fusarium commune* is associated with wilt and root rot diseases in a range of crops: *Acacia koa*, barley, carnation, carrot, Chinese water chestnut (*Eleocharis dulcis*), Douglas-fir, horseradish, maize, peas, rice, soybean, sugarcane, tobacco, tomato, and white pine. This fungus was originally misidentified as *F. oxysporum*; however, it has been resolved within the FOSC and described as a distinct species since 2003 [28,29,30,31,32]. *Fusarium annulatum* is a morphologically and phylogenetically diverse species which has been recently demonstrated as distinct from *F. proliferatum* [19]. *F. annulatum* is a member of the FFSC, which has been recorded as a pathogen on more than 200 plant hosts primarily in subtropical countries [11,19,33,34].

The *rpb2* and *tef1* genes possess high discriminatory power and are well represented in the GenBank database. The *tef1* locus is frequently used as the first choice for taxonomic studies of *Fusarium* due to its single-copy occurrence and high degree of sequence polymorphism among closely related species, while *rpb2* is the second-best gene for discriminating between closely related species (Appendix A) [11,13,16]. In the combined *rpb2* + *tef1* tree, the saffron pathogens were resolved into the three species *F. annulatum*, *F. commune*, and *F. nirenbergiae* with high support values. It is worth mentioning that the *ITS* data were not used in the MLSA, due to their excessive variability within *Fusarium* [10,11,35] and their inability to resolve species.

The fungi isolated in this study from saffron morphologically resembled *F. oxysporum*; however, on close examination, they could be discriminated from each other as *F. annulatum* and *F. commune* based on morphological criteria. Although both species form polyphialides, *F. commune* also produces long, slender monophialides; these microscopic features distinguish *F. annulatum* and *F. commune* from *F nirenbergiae* and *F. oxysporum* [11,13,23,26,28]. Chlamydospores were only absent in *F. annulatum* [19]. The morphological identification validity was corroborated by phylogenetic analysis derived from the molecular data.

We conclude that corm rot is a disease complex, induced by one or more *Fusarium* spp. (*F. annulatum, F. commune*, and *F. nirenbergiaeas*), as observed in several other agricultural crops. As an example, ten putative *Fusarium* species have been associated with yam wilt in China (i.e., *F. asiaticum*, *F. commune*, *F. cugenangense*, *F. curvatum*, *F. gossypinum*, *F. fujikuroi*, *F. nirenbergiae*, *F. odoratissimum*, *F. solani*, and *F. verticillioides*) [15]. Similarly, eight species, including *F. acuminatum*, *F. boothii*, *F. equiseti-incarnatum*, *F. graminearum*, *F. oxysporum*, *F. proliferatum*, *F. solani*, and *F. subglutinans*, have been shown to induce root rot of Zea mays in the USA [36] and three fusarioid species, *F. oxysporum* f. sp. *opuntiarum*, *Fusarium proliferatum*, and *Neocosmospora falciformis*, were found associated with dry rot and soft rot of succulent plants in Iran [37].

In spite of our efforts, we have not yet been successful in establishing species-specific diagnostic features for *F. annulatum*, *F. commune*, and *F. nirenbergiae* which induced rot on saffron plants. Further studies are needed to fully determine the pathogen-specific symptoms of corm rot. Additionally, some rotted corms exhibited slightly different symptoms in terms of severity, intensity, color, or shape, possibly due to secondary infection by saprobic bacteria and fungi or environmental conditions such as humidity or mechanical injury, as has been documented in previous studies [30,35,38,39,40,41].

Strains of *F. annulatum* and *F. commune* are reported as pathogens of bakanae and root rot diseases in rice, respectively [19,28,42,43,44]. Saffron is commonly planted after rice in China as a rotation. This raises the intriguing possibility that rice might serve as a reservoir or alternative host for pathogens of saffron and/or vice versa [19,28,42,43,44]. A first step in investigating that hypothesis will be to investigate the host ranges of these strains: are they able to colonize and/or infect both rice and saffron?

## 5. Conclusions

Overall, data obtained in this study confirm *Fusarium* species are a serious limitation for the commercial production of saffron. Although *F. nirenbergiae* was the prevalent species inciting corm rot in the surveyed areas, *F. annulatum,* and *F. commune* were also recovered from diseased plants, showing to be very aggressive and virulent on saffron. In a disease complex, the frequency of pathogens may vary due to cultivars, agricultural practices, meteorological and climatological parameters, etc. In addition, our survey provides an overview on the biodiversity, distribution, and etiology of *Fusarium* spp. associated with corm rot of *C. sativus* in China, thus enabling the development of better environmentally friendly management strategies.

## Figures and Tables

**Figure 1 jof-08-00515-f001:**
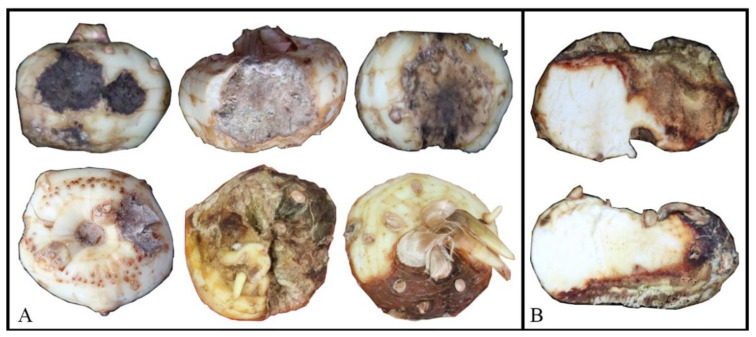
(**A**) Typical field symptom of *Fusarium* corm rot on saffron in China. (**B**) Longitudinal section of corms exhibiting rot developing into endosperm.

**Figure 2 jof-08-00515-f002:**
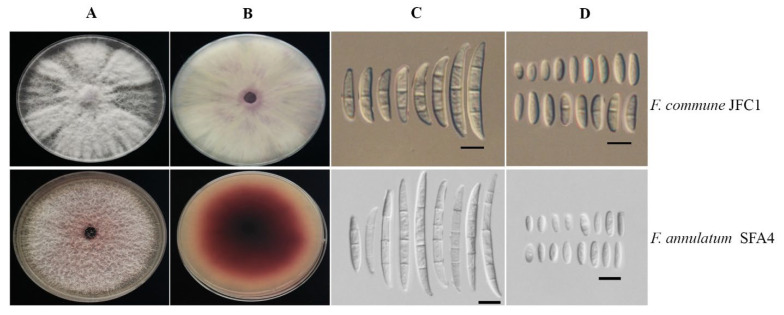
Colony and conidia morphology of *F. commune* and *F. annulatum* isolated from symptomatic corms of Crocus sativus. (**A**) Upper view of a colonies on PDA; (**B**) reverse view of colony on PDA; (**C**) Macroconidia; (**D**) Microconidia—scale bars: (**C**,**D**) = 10 µm.

**Figure 3 jof-08-00515-f003:**
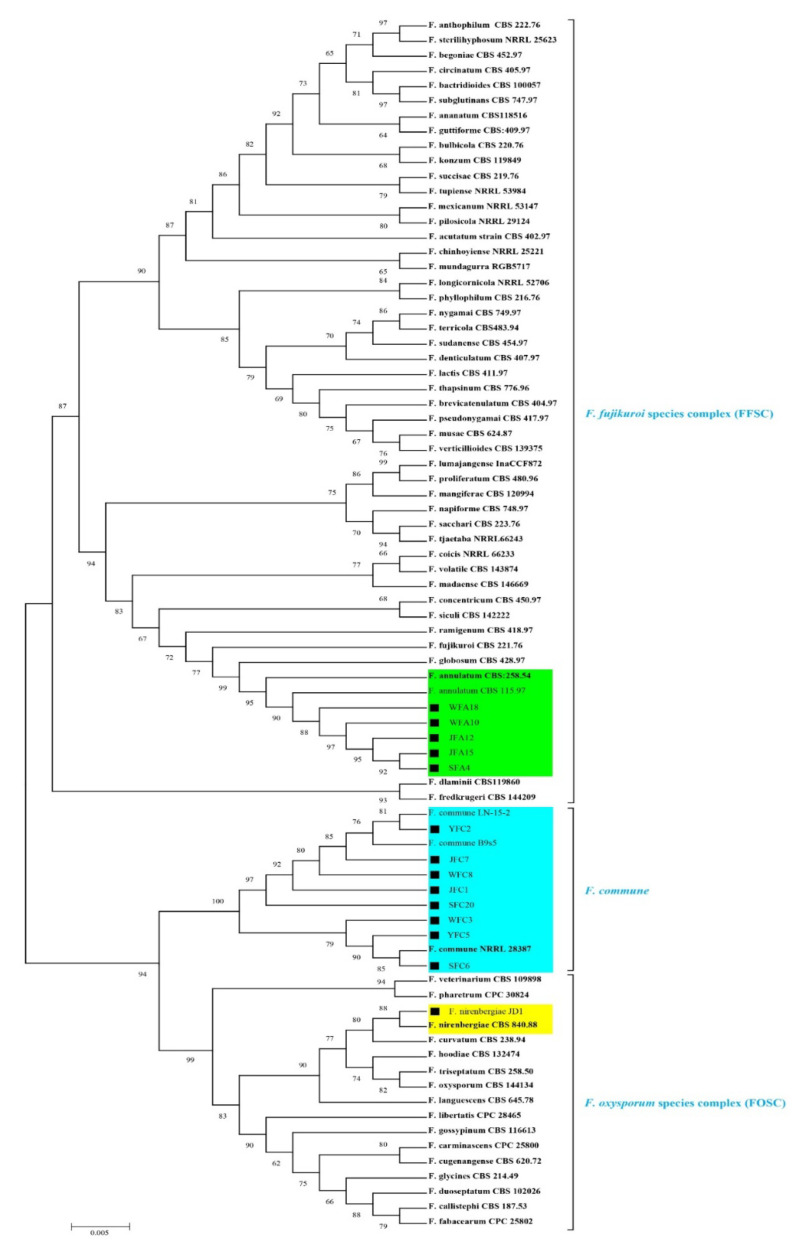
Phylogenetic tree generated from maximum likelihood analysis of combined *rpb2* and *tef1* sequences, depicting the phylogenetic relationships of *Fusarium* species causing corm rot disease in *Crocus sativus* from China. Isolates recovered from saffron during the current study are indicated by a black square (■). Clades including isolates obtained from saffron are shaded in color. Ex-type, neotype, and epitype strains are indicated in bold. Support values representing bootstrap percentages are shown on the branches.

**Figure 4 jof-08-00515-f004:**
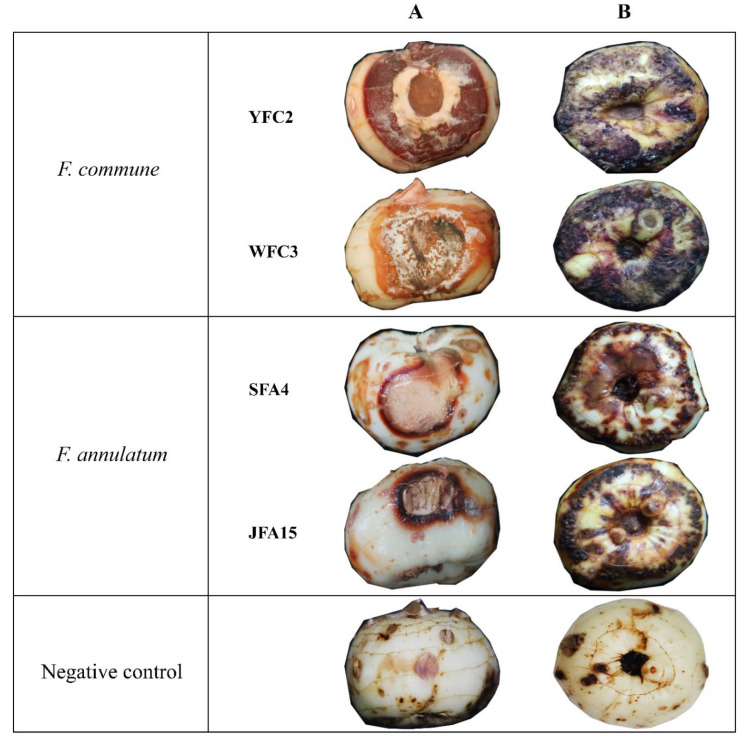
Pathogenicity of *F. commune* and *F. annulatum* isolates on *C. sativus*. (**A**) Corm rot symptoms resulting from inoculation with mycelial plugs. (**B**) Symptoms of rot on corms inoculated with conidial suspensions.

**Table 1 jof-08-00515-t001:** Sampling details, isolates number, and frequency of fungal species identified in the present study.

Province	City	Geographic Coordinates	Sample No.	*F. nirenbergiae* ^a^	*F. commune* ^a^	*F. annulatum* ^a^
Zhejiang	Jiande	119.5998° E 29.2401° N	45	58	26	19
	Wuyi	119.8165° E 28.8926° N	25	37	15	12
Shanghai	Chong Ming Dao	121.3975° E 31.6227° N	25	31	11	8
Yunnan	Shangri-la	99.7060° E 27.8230° N	25	26	18	-
Total	-	-	120	152	70	39
Frequency (%)	-	-	-	58.2	26.8	14.9

^a^ Number of isolates for each species.

**Table 2 jof-08-00515-t002:** Saffron isolates of *Fusarium* spp. from four plantation regions of China used in this study and their GenBank accession numbers.

Origin	Species	Representative Isolate	GenBank Accession Number		
			*ITS*	*tef1*	*rpb2*
Jiande	*F. commune*	JFC1	MZ313313	MZ338563	MZ338571
		JFC7	MZ313314	MZ338564	MZ338572
	*F. annulatum*	JFA12	MZ313130	MZ338578	MZ338573
		JFA15	MZ313131	MZ338579	MZ338574
Chong Ming Dao	*F. commune*	SFC6	MZ318050	MZ338561	MZ338569
		SFC20	MZ318051	MZ338562	MZ338570
	*F. annulatum*	SFA4	MZ313134	MZ338582	MZ338577
Wuyi	*F. commune*	WFC3	MZ313309	MZ338557	MZ338565
		WFC8	MZ313310	MZ338558	MZ338566
	*F. annulatum*	WFA10	MZ313132	MZ338580	MZ338575
		WFA18	MZ313133	MZ338581	MZ338576
Shangri-la	*F. commune*	YFC2	MZ313311	MZ338559	MZ338567
		YFC5	MZ313312	MZ338560	MZ338568

**Table 3 jof-08-00515-t003:** *Fusarium* strains used in this study.

Species	Culture Accession	GenBank Accession	
		*rpb2*	*tef1*
*Fusarium carminascens*	CPC 144738 T	MH484937	MH485028
*F. contaminatum*	CBS 111552 T	MH484901	MH484992
*F. pharetrum*	CBS 144751T	MH484952	MH485043
*F. veterinarium*	CBS 109898 T	MH484899	MH484990
*F. cugenangense*	CBS 620.72	MH484879	MH484970
*F. curvatum*	CBS 238.94 T	MH484893	MH484984
*F. fabacearum*	CPC 25802 T	MH484939	MH485030
*F. glycines*	CBS 144746 T	MH484942	MH485033
*F. gossypinum*	CBS 116613 T	MH484909	MH485000
*F. languescens*	CBS 645.78 T	MH484880	MH484971
*F. libertatis*	CPC 28465 T	MH484944	MH485035
*F. nirenbergiae*	CBS 840.88 T	MH484887	MH484978
	JD1	MT864705	MT814630
	JD2	MT864708	MT814633
	JD3	MT864711	MT814629
	JD4	MT864712	MT814625
	SH1	MT864704	MT814627
	SH2	MT864706	MT814631
	BZ1	MT864700	MT814622
	BZ3	MT864701	MT814623
	BZ4	MT864702	MT814624
	WY5	MT864713	MT814635
	WY9	MT864709	MT814634
	WY11	MT864710	MT814628
	GY2	MT864703	MT814626
	GY6	MT864707	MT814632
*F. oxysporum*	CBS 144134 ET	MH484953	MH485044
*F. hoodiae*	CBS 132474 T	MH484929	MH485020
*F. duoseptatum*	CBS 102026 T	MH484896	MH484987
*F. callistephi*	CBS 187.53 T	MH484875	MH484966
*F. triseptatum*	CBS 258.50 T	MH484910	MH485001
*F. languescens*	CBS 645.78 T	MH484880	MH484971
*F. elaeidis*	CBS 217.49 T	MH484870	MH484961
*F. commune*	NRRL 28387	HM068356	HM057338
	LN-15-2	MH716813	MH716809
	B9s5	MN892350	MK560330
	NRRL 38348	-	FJ985389
	NRRL28058	-	AF324333
	CBS 110090 T	MW934368	-
*F. acutatum*	CBS 402.97 T	MW402768	MW402125
*F. agapanthi*	NRRL 54463 T	KU900625	KU900630
*F. ananatum*	CBS 118516 T	LT996137	LT996091
*F. andiyazi*	CBS 119857 T	LT996138	MN193854
*F. annulatum*	CBS 115.97	MW402785	MW401973
	CBS 258.54 T	MT010983	MT010994
*F. anthophilum*	CBS 222.76 ET	MW402811	MW402114
*F. bactridioides*	CBS 100057 T	MN534235	MN533993
*F. begoniae*	CBS 452.97 T	MN534243	MN533994
*F. brevicatenulatum*	CBS 404.97 T	MN534295	MN533995
*F. bulbicola*	CBS 220.76 T	MW402767	KF466415
*F. xyrophilum*	NRRL 62721 T	MN193905	-
*F. chinhoyiense*	NRRL 25221 T	MN534262	MN534050
*F. subglutinans*	CBS 747.97 NT	MW402773	MW402150
*F. circinatum*	CBS 405.97 T	MN534252	MN533997
*F. coicis*	NRRL 66233 T	KP083274	KP083251
*F. concentricum*	CBS 450.97 T	JF741086	AF160282
*F. denticulatum*	CBS 407.97 T	MN534274	MN534000
*F. dlaminii*	CBS 119860 T	KU171701	MW401995
*F. echinatum*	CBS 146497 T	-	MW834273
*F. fredkrugeri*	CBS 144209 T	LT996147	LT996097
*F. fujikuroi*	CBS 221.76 T	KU604255	MN534010
*F. globosum*	CBS 428.97 T	KF466406	KF466417
*F. guttiforme*	CBS 409.97 T	MT010967	MT010999
*F. konzum*	CBS 119849 T	MW402733	LT996098
*F. lactis*	CBS 411.97 ET	MN534275	MN193862
*F. longicornicola*	NRRL 52706 T	JF741114	JF740788
*F. lumajangense*	InaCCF872 T	LS479850	LS479441
*F. madaense*	CBS 146669 T	MW402764	MW402098
*F. mangiferae*	CBS 120994 T	MN534271	MN534017
*F. mexicanum*	NRRL 53147 T	MN724973	GU737282
*F. mundagurra*	RGB5717 T	KP083276	KP083256
*F. musae*	CBS 624.87 T	MW402772	FN552086
*F. napiforme*	CBS 748.97 T	MN534291	MN193863
*F. nygamai*	CBS 749.97 T	EF470114	MW402151
*F. ophioides*	CBS 118512 T	MN534303	MN534022
*F. phyllophilum*	CBS 216.76 T	KF466410	MN193864
*F. pilosicola*	NRRL 29124 T	MN534248	MN534055
*F. proliferatum*	CBS 480.96 ET	MN534272	MN534059
*F. pseudonygamai*	CBS 417.97 T	MN534285	AF160263
*F. ramigenum*	CBS 418.97 T	KF466412	KF466423
*F. sacchari*	CBS 223.76 ET	JX171580	MW402115
*F. siculi*	CBS 142222 T	LT746327	LT746214
*F. succisae*	CBS 219.76 ET	MW402766	AF160291
*F. sudanense*	CBS 454.97 T	MN534278	MN534037
*F. terricola*	CBS 483.94 T	LT996156	MN534042
*F.thapsinum*	CBS 776.96 T	MN534289	MN534044
*F. tjaetaba*	NRRL 66243 T	KP083275	KP083263
*F. verticillioides*	CBS 139375 T	MW402802	MW402068
*F. volatile*	CBS 143874 T	LR596006	LR596007
*F. werrikimbe*	CBS 125535 T	MN534304	MW928846
*F. prieskaense*	CBS 146498 T	-	MW834275
*F. pseudoanthophilum*	CBS 414.97 T	MT010980	MT011006
*F. ficicrescens*	CBS 125178 T	KT154002	MT011004
*F. pseudocircinatum*	CBS 449.97 T	MT010968	MT011003
*F. foetens*	CBS 110286 T	MW928825	MT011001
*F. pininemorale*	CMW 25243 T	MN534250	MN534026
*F. inflexum*	NRRL 20433 T	JX171583	AF008479
*F. sterilihyphosum*	NRRL 25623 T	MN193897	MN193869
*F. xylarioides*	NRRL 25486 T	-	AY707136
*F. hostae*	NRRL 29889 T	JX171640	-
*F. ramigenum*	CBS 418.97 T	MT010975	MT011012
*F. redolens*	NRRL 25600 T	MT409443	-
*F. udum*	BBA 65058 T	KY498875	-

CBS: Westerdijk Fungal Biodiverity Institute (WI), Utrecht, The Netherlands. NRRL (Northern Regional Research Laboratory): Agricultural Research Service Culture Collection Database, Peoria, USA. CMW: the working collection of FABI (Forestry and Agricultural Biotechnology Institute), University of Pretoria, South Africa. BBA: Julius Kühn Institute, Institute for Epidemiology and Pathogen Diagnostics, Berlin & Braunschweig, Germany. CPC: Collection of P.W. Crous. T: Ex-type specimen. NT: neotype specimen. ET: Ex-epitype specimen.

## Data Availability

Sequences have been deposited in GenBank (Table 2). The data presented in this study are openly available in NCBI. Publicly available datasets were analyzed in this study. These data can be found here: https://www.ncbi.nlm.nih.gov/ (accessed on 15 August 2021).

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
