# Peer review of "Molecular and Pathogenic Characterization of Fusarium Species Associated with Corm Rot Disease in Saffron from China"

_jof, 2022, doi:10.3390/jof8050515_

Round 1
Reviewer 1 Report
After careful evaluation of the article title “Molecular and Pathogenic Characterization of Fusarium Species Associated with Corm Rot Disease in Saffron from China”, you can find below my comments.
Introduction
- Please state if saffron is an important crop in China and how much area it covers
- Please indicate any more pathogens found on saffron corms (except Fusarium spp.)
- Please note that there are non-pathogenic Fusarium races that are used in plant protection
Materials and Methods
2.3. Pathogenicity Studies
- Please provide information why 13 representative isolates of Fusarium species were selected based on what traits or at random?
Conclusions
- Please expand on whether the discovered species of Fusarium may dominate in the future in the cultivation of saffron, which means for the broadly understood protection of this plant and economic profitability
Yours sincerely
Author Response
Dear Professor,
We sincerely express our deepest gratitude for your comments and gave us this chance to publish our paper in the Journal of Fungi. We have made all changes based on your comments. We used the Track Changes option to record all the updates in the manuscript.
We hope the revised manuscript is now suitable for publication.
Regards
Ali
P.S. Kindly see the following:
Introduction
- Please state if saffron is and how much area it covers
Our response: This plant is widely used as an important herbal medicine crop in China with a total cultivating area of nearly 600 hectares. It is very expensive.
- Please indicate any more pathogens found on saffron corms (except Fusarium spp.)
Dear Professor, Did you mean that I should put the following paragraph at the line 39 in first page?
“Various pathogens affecting corm including Aspergillus niger, A. terreus, A. flavus, A. flavipes, Bacillus croci, Burkholderia gladioli, Fusarium spp., Macrophomina phaseolina, Mucor sp., Penicillium solitum, P. cyclopium, P. corymbiferum, Phoma crocophila, Pseudomonas gladioli, Pythium spp., Rhizoctonia crocorum, R. violacea, Rhizopus nigricans, Sclerotium rolfsii, Stromatinia gladioli, and Uromyces croci”.
- Please note that there are non-pathogenic Fusarium races that are used in plant protection
Our response: Thnak you for your suggestion. We have added this pharase “ However, there are non-pathogenic strains of Fusarium that are used in plant protection”
- Please provide information why 13 representative isolates of Fusarium species were selected based on what traits or at random?
Our response: Thank you for the good question. To reduce the number of isolates for sequencing, phylogenetic analyses, and pathogenicity tests, 13 representative isolates were selected based on geographical zones, colony morphological characteristics, and tef1 & ITS sequences. Furthermore, we have checked all collected isolates (including representative isolates) with ISSR fingerprinting patterns to verify our work accuracy (data not shown).
- Please rewrite the conclusion. Please expand on whether the discovered species of Fusarium may dominate in the future in the cultivation of saffron, which means for the broadly understood protection of this plant and economic profitability
Our response: We have revised the conclusion. Furthermore, It is common in a complex disease the prevalency and isolation frequency of pathogens may vary or gradually change due to cultivars, farming system, environmental conditions, and so on. In other words, a species may have low prevalency and aggressiveness in a special geographical region or on a special cultivar, but this species can be more aggressive and prevalent in another geographical region or on another cultivar.
(References: Wilson 2002; Balci et al. 2007; Guo et al. 2014; Baroncelli, R. et al. 2015; Hay et al. 2015; Wang, Yu-Chun, et al. 2016; Da Lio, Daniele, et al. 2018; Robinson et al. 2019; Hong et al. 2020; Khodadadi et al. 2020; Nishad and Ahmed 2020; Darapanit et al. 2021; Dobbs et al. 2021; Maymon et al 2021; Wang, Wei, et al. 2021).
Reviewer 2 Report
Well written paper on molecular identification of Fusarium species causing saffron rot. Altough saffron in not very popular crop, metedology of Fusarium identification can be very useful for other Fusarium researchers.
Only one editorial comment: authors should use italics for Latin names of species as well as gene names.

Author Response
Dear Professor
Thank you very much for your support
We have changed to Italics Style all of the Latin names of species as well as gene names.
Regard
Ali
Reviewer 3 Report
Thank you for the opportunity to review this article.
I believe that the manuscript has high level, is clearly written and well organized, and I am in favor of publishing, after minor revision.
Some minor remarks:
Please write all pathogens in italics.
Line 123.in vitro... please write in italics.
Please rewrite the conclusions.
Author Response
Dear Professor
I am grateful for your support and comments.
I have rewritten the conclusions
Kind Regards
Ali